# 3D NAND Flash Memory Based on Double-Layer NC-Si Floating Gate with High Density of Multilevel Storage

**DOI:** 10.3390/nano12142459

**Published:** 2022-07-18

**Authors:** Xinyue Yu, Zhongyuan Ma, Zixiao Shen, Wei Li, Kunji Chen, Jun Xu, Ling Xu

**Affiliations:** 1School of Electronic Science and Engineering, Nanjing University, Nanjing 210093, China; yuxinyue2083@126.com (X.Y.); 18932366599@163.com (Z.S.); weili@nju.edu.cn (W.L.); kjchen@nju.edu.cn (K.C.); junxun@nju.edu.cn (J.X.); lingxu@nju.edu.cn (L.X.); 2Collaborative Innovation Center of Advanced Microstructures, Nanjing University, Nanjing 210093, China; 3Jiangsu Provincial Key Laboratory of Photonic and Electronic Materials Sciences and Technology, Nanjing University, Nanjing 210093, China

**Keywords:** 3D flash memory, nanocrystalline Si, multilevel storage

## Abstract

As a strong candidate for computing in memory, 3D NAND flash memory has attracted great attention due to the high computing efficiency, which outperforms the conventional von-Neumann architecture. To ensure 3D NAND flash memory is truly integrated in the computing in a memory chip, a new candidate with high density and a large on/off current ratio is now urgently needed. Here, we first report that 3D NAND flash memory with a high density of multilevel storage can be realized in a double-layered Si quantum dot floating-gate MOS structure. The largest capacitance–voltage (C-V) memory window of 6.6 V is twice as much as that of the device with single-layer nc-Si quantum dots. Furthermore, the stable memory window of 5.5 V can be kept after the retention time of 10^5^ s. The obvious conductance–voltage (G-V) peaks related to the charging process can be observed, which further confirms that the multilevel storage can be realized in double-layer Si quantum dots. Moreover, the on/off ratio of 3D NAND flash memory with a nc-Si floating gate can reach 10^4^, displaying the characteristic of a depletion working mode of an N-type channel. The memory window of 3 V can be maintained after 10^5^ P/E cycles. The programming and erasing speed can arrive at 100 µs under the bias of +7 V and −7 V. Our introduction of double-layer Si quantum dots in 3D NAND float gating memory supplies a new way to the realization of computing in memory.

## 1. Introduction

With the era of big data and artificial intelligence approaching, computing in memory has attracted great attention due to the high computing efficiency, which outperforms the conventional von-Neumann architecture. As a strong candidate of computing in memory, Si-based 3D NAND flash with high density, a large on/off current ratio, and perfect compatibility with CMOS technology is in high demand [1,2,3]. Among the candidates of 3D NAND flash memory, nanocrystalline silicon (nc-Si) floating-gate memory is considered to be promising due to its low power, fast program and erase speed, and high durability [4,5,6,7,8]. In particular, its high compatibility with modern microelectronics technology is beneficial to be integrated with computing in a memory chip. To make nc-Si floating-gate memory truly useable in 3D NAND flash memory, the increase in the injection charge density and the retention characteristic has been always a concern of the industry [7,8]. As for the research progress of floating-gate memory based on Ge nanocrystalline dots, R. Bar et al. reported that the memory window of multilayer Ge nanocrystalline floating-gate memory increases from 4.4 to 6.4 V as the number of Ge nanocrystal layers increases from 1 to 5. Additionally, the retention time of Ge nanocrystalline floating-gate memory with five layers reaches 10^4^ s [9]. According to the report by C. Palade et al., the bilayer Ge nanocrystal floating-gate memory has the largest memory window of 6.1 V [10]. As for the research progress of floating-gate memory based on Si nanocrystalline dots, S.-W Fu et al. reported that the memory window of multilayer Si nanocrystal floating-gate memory increases from 16 to 25.6 V after the density of Si nanocrystal increases from 1.6 × 10^12^ to 2.6 × 10^12^ [11]. However, the capacitance reduces sharply after the retention time reaches 10^3^ s. As reported by V. Turchanikov, a single- and double-nanocrystal floating-gate layer can achieve a memory window of 1 V and 2 V, respectively [12]. According to the report by K. Ilse [13], the memory window of floating-gate memory with five layers of Si nanoparticles embedded in Al_2_O_3_ is only 0.7 V, which is reduced to 0.5 V after a retention time of 10^5^ s. Contrasting with the memory window and retention time of the floating-gate memory based on Ge and Si nanocrystalline dots, the memory window of our double nc-Si floating gate memory can reach the maximum value of 6.6 V and a stable memory window of 5.5 V can be kept after a retention time of 10^5^ s, which reveals that our device has the potential to be applied in 3D flash memory. It is worth noting that a double nc-Si storage layer was less reported in 3D NAND floating-gate memory, which supplies a new way to the realization of computing in memory.

In this paper, 3D NAND flash memory was obtained by using a double-layered nc-Si quantum dots MOS structure as the floating gate. It is found that the capacitance–voltage (C-V) memory window can be 6.6 V when the electron and hole injection reaches saturation, which is about twice as much as that of the device with single-layer nc-Si quantum dots. Furthermore, the larger memory window of 5.5 V can be kept after a retention time of 10^5^ s. The analysis of the conductance–voltage (G-V) curve reveals that a double charging peak can be detected during the charging process, which further confirms that charge is stored in double-layer Si quantum dots. Moreover, the on/off ratio of 3D NAND flash memory with a nc-Si floating gate can reach 10^4^, displaying the characteristic of a depletion working mode of an N-type channel. The memory window of 3 V can be retained after 10^5^ P/E cycles. The programming and erasing speed can reach 100 µs under the bias of +7 V and −7 V.

## 2. Materials and Methods

Figure 1a shows the schematic diagram of 3D flash memory based on junctionless a-Si:H channels with a double-layered nc-Si dots floating gate structure. P-type (100) silicon wafers were used as the substrate. The resistivity of the Si substrate was 6~9 Ω cm. For the MOS structure, the Si substrate was cleaned using a standard RCA process followed by 4% diluted HF to remove the native oxide. Then, the first tunnel oxide layer of 9 nm was obtained by thermal oxidation of Si at 850 °C, which was followed by the first amorphous silicon layer of 4 nm grown at a temperature of 250 °C in the PECVD chamber. The second tunnel oxide layer of 3 nm was deposited by using N_2_O and SiH_4_ as the precursor. Subsequently, the second amorphous silicon layer of 4 nm was deposited at a temperature of 250 °C in the PECVD chamber. Additionally, the control nitride layer of 25 nm was grown at the top of the second a-Si layer by decomposition of SiH_4_ and NH_3_. Finally, the sample was thermally annealed at 1000 °C under N_2_ ambient for half an hour. For comparison, a reference sample with the same thickness of a single nc-Si layer was fabricated by a similar process. For the electric measurements, aluminum (Al) top electrodes were fabricated on the surfaces of the samples by thermal evaporation with a shadow mask to form a circular spot. Al was deposited at the Si substrate as the bottom electrode using the thermal evaporation system. The top and the back electrode were alloyed at 400 °C to form ohmic contact. To obtain 3D NAND flash memory, as shown in Figure 1a, a silicon oxide insulating layer of 300 nm was grown on the surface of the Si substrate by using the wet oxidation method. Two layers of amorphous Si:H and SiO_2_ layer were sequentially deposited on the surface of the SiO_2_ insulating layer in the PECVD system at 300 °C. The thickness of the Si:H and SiO_2_ layer was 100 nm and 60 nm. Then, the two layers of a-Si:H and SiO_2_ were patterned by the electron beam lithography and dry-etched to form the amorphous silicon channel. We prepared a double-layer nc-Si floating-gate on the surface of double-layer a-Si channels according to the preparation method of the MOS structure. In the following process, an a-Si:H terrace was obtained by etching to prepare for the drain electrode. The hole of the drain electrode was formed by the etching of each a-Si:H terrace, which was patterned by photolithography. Additionally, the position of the drain electrode in each a-Si:H was different from each other. Meanwhile, the hole of the source electrode was obtained by etching a-Si:H from the first to the second a-Si:H layer to expose two a-Si:H layers. Finally, all the electrode holes were filled by the evaporation of aluminum to form source electrodes and drain electrodes. Additionally, the metal of the two gate electrodes was also deposited by thermal evaporation followed by lift-off technology to prepare for electrical measurement. The microstructure of the annealed samples was revealed by high-resolution cross-section transmission electron microscopy (HRTEM) with a JEM2010 electron microscope working at 200 kV. The C-V, G-V, transfer, as well as output characteristic were measured by using an Agilent B1500A at room temperature.

## 3. Results and Discussion

The schematic diagram of 3D flash memory based on double-layer a-Si:H channels with a double-layered nc-Si dots floating gate structure is displayed in Figure 1a. Additionally, the top-view morphology of the 3D flash memory with a channel width of 0.2 μm can be observed in Figure 1b. The schematic diagram of the MOS device with a double-layer nc-Si floating gate for C-V measurement is shown in Figure 1c. During the C-V measurement, the bottom aluminum electrode is grounded and the top aluminum electrode is applied with varying gate voltage. From the HRTEM image of the double-layer nc-Si dot floating-gate MOS structure, as presented in Figure 1d, it can be clearly seen that there are perfect crystals of less than 3.5 nm distributed in two a-Si sublayers. The (100) crystallographic orientation of nc-Si in the first and the second sublayer is evident. It can be observed that the nc-Si dots are separated by amorphous silicon in the first nc-Si layer and in the second nc-Si layer. The thicknesses of the two tunneling oxide layers are 9 nm and 2.5 nm, respectively.

Figure 2a,b present the C-V characteristic curves of the annealed double-layer nc-Si device and the annealed single-layer nc-Si device, respectively. The frequency is 1 MHz and the step amplitude is 20 mV. The bias scanning direction ranges from the charge accumulation region to the charge inversion region, and then starting from the charge inversion region to the charge accumulation region. It can be observed that the hysteresis window of the double-layer device reaches 6.6 V when the saturated bias ranges from −11 V to 10 V, while the memory window of the single-layer device is only 2.9 V when the saturated bias increases from −9 V to 8 V. This indicates that the memory density of the double-layer device is higher than that of the single-layer device. It is also noticed that the memory window under the positive voltage is larger than that under the negative voltage for both the double-layer nc-Si device and the single-layer nc-Si device. This means that the number of the electron injection is higher than that of the hole injection, because the efficient mass of the hole is larger than that of the electrons. Thus, the electron injection is easier than that of the holes.

Figure 3a shows the change in flat-band voltage with gate voltage, with the scanning voltage increasing to 7 V. The electron and the hole injection of the single-layer device reach saturation, respectively. The hysteresis window of the single-layer device is always larger than that of the double-layer device before saturation. The electron and hole injections of the single-layer device are more than those of the double-layer device. After the saturation, the flat-band voltage of the single-layer device does not move any more, and the hysteresis window of the double-layer device continues to increase at −10 V and 9 V; the injection of holes and electrons reaches saturation. The saturation window of the double-layer structure is more than twice that of the single-layer structure. It indicates that the surface charge density is more than twice that of the single-layer structure. Considering the Coulomb-blocking effect in nc-Si quantum dots, only one charge can be stored in one nc-Si dot, which makes the second charge difficult to be injected into the nc-Si dot. Therefore, the equivalent surface density of double-layer nc-Si is twice that of single-layer nc-Si. In addition, the existence of tunneling oxide leads to the introduction of new interface states where a part of charges can be stored. So, the surface charge density of a double-layer nc-Si device will be more than twice as much as that of a single-layer nc-Si device. As displayed in Figure 3b, the retention characteristics of the nc-Si floating-gate MOS structure with single-layer nc-Si and double-layer nc-Si was tested under the pulses of ±10 V/s and ±8 V/s, respectively. With the time increasing, the leakage of electrons and holes causes the flat-band voltage to shift to the initial state. For the device with single-layer nc-Si, we can observe that the flat-band voltage decreases with time increasing. The hysteresis window of the device with single-layer nc-Si is about 1.5 V after 10^5^ s. The hysteresis window of the device with double-layer nc-Si remains unchanged even after 5 × 10^3^ s. As displayed in Figure 4, the energy band diagram of the double-layer nc-Si quantum dot floating gate structure shows that the reduction in the leakage rate is due to the barrier formed by the first layer of nc-Si and the second tunneling layer. Although the leakage rate of the first nc-Si is faster than that of the second layer, the total leakage rate of the double-layer nc-Si device is slower than that of the single-layer nc-Si device.

The G-V and C-V characteristics of double-layer nc-Si floating-gate devices at different frequency of 500 KHZ, 700 KHZ, and 900 KHZ are shown in Figure 5a–c. A conductance peak at −1.6 V with a shoulder is detected, which is located at the depletion region and the accumulation region of the C-V curve, respectively. In order to further explore the origin of the conductance peak and the shoulder, we performed Gauss fitting for the G-V curves from 0 V to −4 V with three frequencies as shown in Figure 5d–f. It is shown that the intensity of the conductance peak and the shoulder increase with frequency because more charge traps were unable to respond to the ac-signal change at higher frequencies, which results in greater energy loss [14,15,16]. The conductance peak and the shoulder were well separated to peak 1, peak 2, and peak 3 by Gaussian fitting. With the increase in frequency, peak 1 and peak 2 have little shift, while peak 3 shifts negatively (~0.14 V). According to the report by S. Oda [17,18], peak 1 and peak 2 are related to nc-Si, while peak 3 is a typical response from the interface traps. Compared with the C-V characteristic, it is clear that the three fitted peaks—peak 1 (corresponding to the first-layer floating-gate nc-Si), peak 2 (corresponding to the second-layer floating-gate nc-Si), and peak 3 (corresponding to the interface state)—were located in the depletion region, weak depletion region, and accumulation region, respectively. This indicates that both the first- and the second-layer nc-Si floating gate played the role of storage charge.

The transfer characteristic of the 3D NAND nc-Si floating-gate memory unit based on a a-Si:H channel with a width of 0.2 μm is shown in Figure 6a. The I_s-d_ (current of source and drain) increases with the gate bias varying from negative to positive, displaying the characteristic of a depletion working mode of an N-type channel. The on/off current ratio can reach 5 × 10^4^. As displayed in Figure 6b, the corresponding output current is enhanced with the positive gate voltage increasing, which further indicates that the conductive type of the a-Si channel is N-type. It is worth noting that the larger memory window of 3 V can be obtained under the programming and erasing bias of +7 V and −7 V. The programming and erasing speed is 100 µs. After 10^5^ P/E cycles, the stable memory window of 3 V can be retained as shown in Figure 6c,d.

## 4. Conclusions

In summary, we successfully obtained 3D NAND flash memory based on a double-layered Si quantum dot floating-gate structure with high density of multilevel storage. The capacitance–voltage (C-V) memory window can reach 6.6 V, which is twice as much as that of the device with single-layer nc-Si quantum dots. The stable memory window of 5.5 V can be kept after a retention time of 10^5^ s. The analysis of the conductance–voltage (G-V) curve reveals that charge storage can be realized in each layer of the double-layer nc-Si dots. Moreover, the current on/off ratio of 3D NAND flash memory with a nc-Si floating gate can reach 10^4^, displaying the characteristic of a depletion working mode of an N-type channel. The MOSFET memory window of 3 V can be retained after 10^5^ P/E cycles. The programming and erasing speed is 100 µs under the bias of +7 V and −7 V. Our introduction of double-layered Si quantum dots in 3D NAND floating-gate memory is of great significance to the realization of computing in memory.

## Figures and Tables

**Figure 1 nanomaterials-12-02459-f001:**
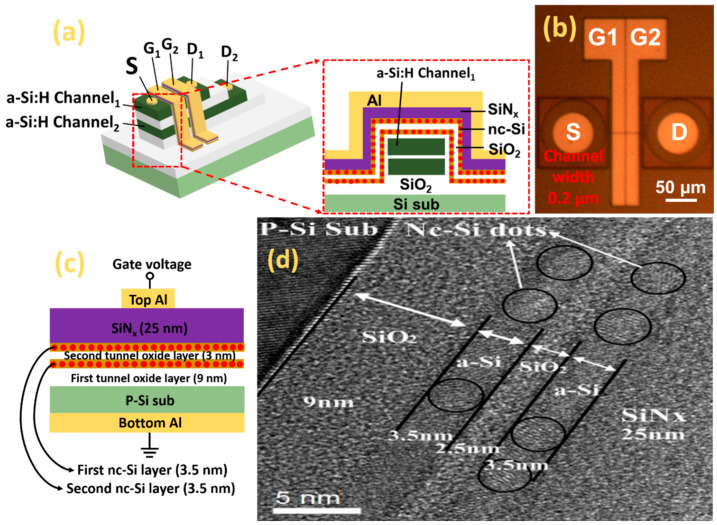
(**a**) Schematic diagram of 3D flash memory based on double-layer a-Si:H channels with a double-layered nc-Si dots floating gate structure. (**b**) The top-view morphology of the 3D flash memory with channel width of 0.2 μm. (**c**) Schematic diagram of MOS device with double-layer nc-Si floating gate for C-V measurement. (**d**) HRTEM photo of the SiN_x_/nc-Si/SiO_2_/nc-Si/SiO_2_ double-layered MOS structure. The black circles represent nc-Si quantum dots.

**Figure 2 nanomaterials-12-02459-f002:**
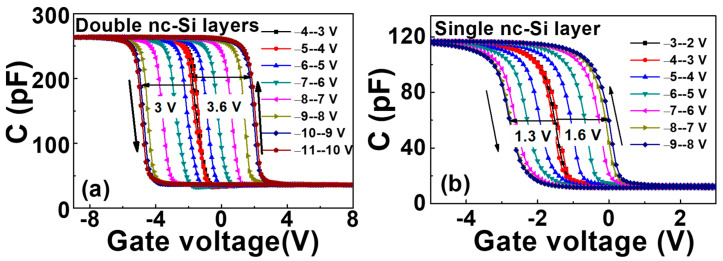
C-V diagrams of (**a**) double-layer nc-Si floating gate memory device and (**b**) single-layer nc-Si floating-gate memory device at 1 MHz.

**Figure 3 nanomaterials-12-02459-f003:**
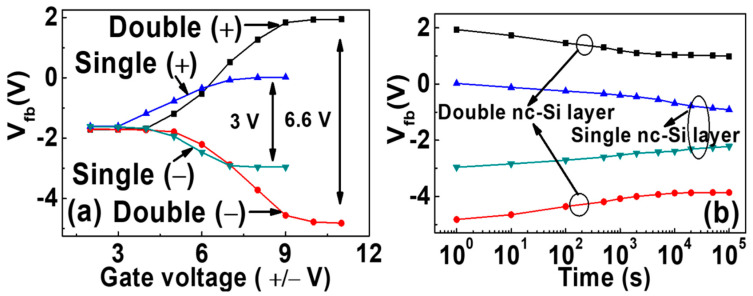
Flat-band voltage variation and the retention characteristics of the nc-Si floating-gate MOS structure with (**a**) single-layer nc-Si and (**b**) double-layer nc-Si.

**Figure 4 nanomaterials-12-02459-f004:**
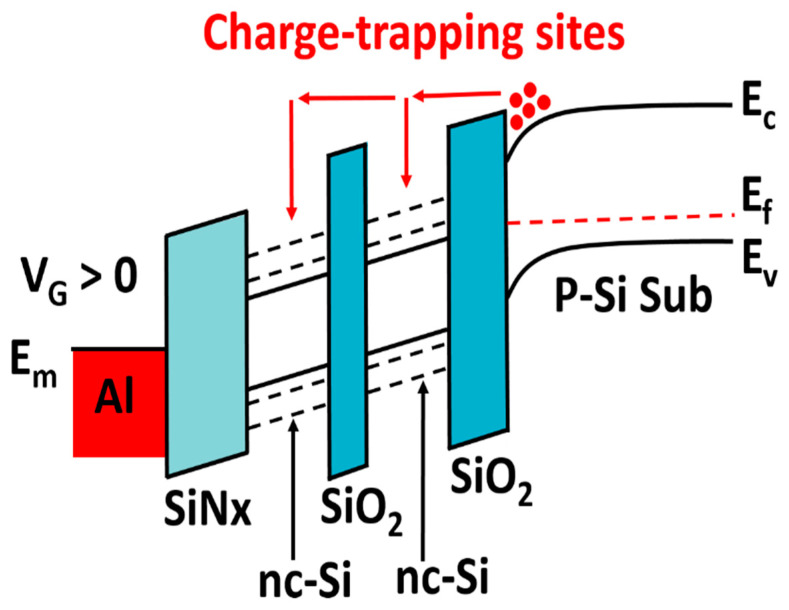
Energy band diagram of double-layer quantum dot floating-gate structure.

**Figure 5 nanomaterials-12-02459-f005:**
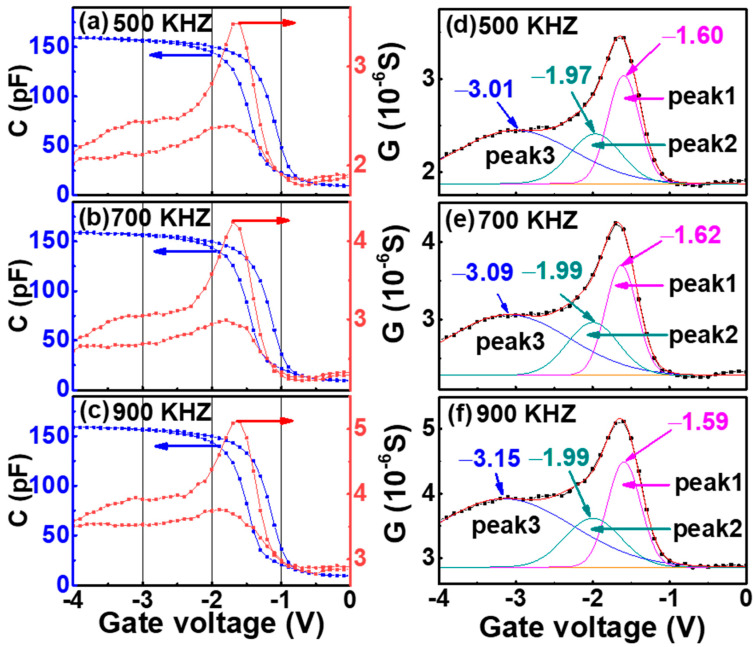
The G-V and C-V characteristic of the double-layer nc-Si floating-gate MOS structure under the frequency of (**a**) 500 KHZ, (**b**) 700 KHZ, and (**c**) 900 KHZ. The G-V curve is fitted by three Gaussian peaks of the annealed double-layer device at (**d**) 500 KHZ, (**e**) 700 KHZ, and (**f**) 900 KHZ. The blue lines represent the C-V curves, the red lines represent the G-V curves and the black lines represent the base line.

**Figure 6 nanomaterials-12-02459-f006:**
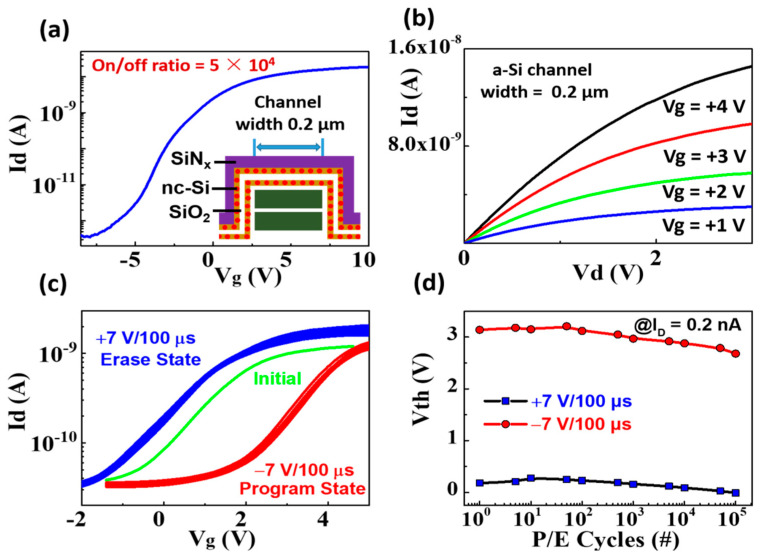
(**a**) The transfer characteristics of the 3D flash memory with the channel width of 0.2 μm (Vd = +1 V), (**b**) the output characteristics of the 3D flash memory with the channel width of 0.2 μm, (**c**) the transfer characteristics after programming and erasing operations at +7 V and −7 V with 100 µs pulse, (**d**) endurance characteristics during the 105 P/E cycles operations at +7 V and −7 V with 100 µs pulse.

## Data Availability

Data can be available upon request from the authors.

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
