# Peer review of "3D NAND Flash Memory Based on Double-Layer NC-Si Floating Gate with High Density of Multilevel Storage"

_nanomaterials, 2022, doi:10.3390/nano12142459_

Round 1

Reviewer 1 Report

The multilayer floating gate is a good approach for enhancing the memory properties, as it “ensures better control of variable programming and retention by multiple tunneling in the floating gate and reduced leakage in the uppermost layer by Coulomb blockade and leakage decrease with the increase of layers number [R. Bar et al., Appl. Phys. Lett. 107, 093102 (2015)]”.

I have the following comments and suggestions attached.

Reviewer 2 Report

The manuscript technically sounds, but the introduction does not support the rest of the work. In particular, it completely lacks of a comparison with the existing literature. In particular, in page 2 (rows 40-42) the authors state that: 

"Most of nc-Si .... is based on single-barrier of single-layer storage ...." 

"Up to now, the introduction of double layer .... is less reported" 

As you can see, the statements are vague and do not help the reader which wonder about the possible existence of a few similar solutions in the literature. I recommend to expand the introduction to highlight the paper novelty in a clear way. 

Reviewer 3 Report

This research shows good innovation and the paper is well written. This work shows good application value in the research of memory devices. Therefore, I recommend that this paper can be accepted for publication after appropriate revisions:

- the experimental part should give a detailed device preparation and characterization process eg. to reach the structure shown in Figure 1 (a) would require some technological processes (perhaps photolithography?, reactive ion etching?). The authors must insert into manuscript more details about necessary technological stages to obtain such a structure.

- before the bottom deposition of Al, was the native oxide removed from the silicon surface?

- I suggest that the author should add/give a complete schematic diagram of the device structure in order to be clear on how the structure is investigated from an electrical point of view, please include in the manuscript a complete structure with the position of the Al electrodes;

- hydrogenation of silicon is not described, please describe it;

-  figures 6 c and d are not cited in the text;

- the journal template is not respected:

e.g. # the requirement is Figure.... to appear in the text and you use "figures" or "Fig.";  #figures 2 and 3 are elongated vertically, please comply with the requirements of the journal regarding the correct image dimensions;

#for figures 2 and 3 please reformulate it so that points (a) and (b) to appear correctly in the figure caption description e.g C-V diagrams of  (a)  double-layer .... and (b) single-layer nc-Si .....

Round 2

Reviewer 3 Report

In the revised manuscript, the author considered the proposed recommendations and brought the appropriate additions. Therefore, I recommend that this paper can be accepted for publication.